From tameness to wariness: chemical recognition of snake predators by lizards in a Mediterranean island

Mencía Abraham
Ortega Zaida zaidaortega@usal.es
Pérez-Mellado Valentín
Department of Animal Biology, University of Salamanca , Salamanca , Spain
Wink Michael
Electronic publication date: 2017 Jan 17
Publication date: 2017
Volume: 5
Electronic Location ID: e2828
Received 2016 Aug 6; Accepted 2016 Nov 23
Copyright: ©2017 Mencía et al.
Copyright year: 2017
Copyright holder: Mencía et al.
License: This is an open access article distributed under the terms of the Creative Commons Attribution License, which permits unrestricted use, distribution, reproduction and adaptation in any medium and for any purpose provided that it is properly attributed. For attribution, the original author(s), title, publication source (PeerJ) and either DOI or URL of the article must be cited.
License URL: https://creativecommons.org/licenses/by/4.0/

Keywords: Insular tameness, Antipredatory behaviour, Islands, Behavioural adaptations, Biological invasions, Lizards, Snakes, Chemoreception, Chemical cues, Mediterranean islands

Funding: University of Salamanca (FPI program) Spanish Ministry of Science and Innovation CGL2012-39850-CO2-02 Financial support was provided to ZO and AM by predoctoral grants of the University of Salamanca (FPI program). This work was also supported by the research project CGL2012-39850-CO2-02 from the Spanish Ministry of Science and Innovation. The funders had no role in study design, data collection and analysis, decision to publish, or preparation of the manuscript.

==============================
Antipredatory defenses are maintained when benefit exceeds cost. A weak predation pressure may lead insular lizards to tameness. Podarcis lilfordi exhibits a high degree of insular tameness, which may explain its extinction from the main island of Menorca when humans introduced predators. There are three species of lizards in Menorca: the native P. lilfordi, only on the surrounding islets, and two introduced lizards in the main island, Scelarcis perspicillata and Podarcis siculus. In addition, there are three species of snakes, all introduced: one non-saurophagous (Natrix maura), one potentially non-saurophagous (Rhinechis scalaris) and one saurophagous (Macroprotodon mauritanicus). We studied the reaction to snake chemical cues in five populations: (1) P. lilfordi of Colom, (2) P. lilfordi of Aire, (3) P. lilfordi of Binicodrell, (4) S. perspicillata, and (5) P. siculus, ordered by increasing level of predation pressure. The three snakes are present in the main island, while only R. scalaris is present in Colom islet, Aire and Binicodrell being snake-free islets. We aimed to assess the relationship between predation pressure and the degree of insular tameness regarding scent recognition. We hypothesized that P. lilfordi should show the highest degree of tameness, S. perspicillata should show intermediate responses, and P. siculus should show the highest wariness. Results are clear: neither P. lilfordi nor S. perspicillata recognize any of the snakes, while P. siculus recognizes the scent of M. mauritanicus and reacts to it with typical well-defined antipredatory behaviours as tail waving and slow motion. These results rise questions about the loss of chemical recognition of predators during island tameness and its related costs and benefits for lizards of insular habitats. In addition, this highlights the necessity for strong conservation measures to avoid the introduction of alien predators.

Introduction

Predation is one of the main evolutionary forces for animals. Animals whose physiological, morphological or behavioural defenses allow them to avoid predators will enhance survival (Endler, 1986; Lima & Dill, 1990). The hunting mechanisms of predators and the defenses of prey usually coevolve in a cost-benefit model (Greene, 1988; Lima & Dill, 1990; Vermeij, 1994; Sih et al., 2010). Therefore some antipredatory adaptations may be costly under low predation pressure, reducing opportunities for feeding or mating (Relyea, 2002; Brönmark, Lakowitz & Hollander, 2011). Insular populations often experience much weaker predation pressures than continental animals, which results in the evolution of tameness, a reduction of antipredatory responses (Darwin, 1839; Curio, 1976; Blumstein & Daniel, 2005). Thus, tameness is a reduction of different behavioural responses and reflects the relaxation of defensive mechanisms. Insular tameness arises when the cost of maintaining the physiological, morphological or behavioural antipredator defences exceeds the benefits in predator-free environments (McNab, 1994; Van Damme & Castilla, 1996; Magurran, 1999; Blumstein & Daniel, 2005; Rödl et al., 2007). Then, the introduction of non-native predators on islands may give rise to novel predator–prey interactions (Strauss, Lau & Carroll, 2006; Sih et al., 2010; Simberloff et al., 2013). Occasionally, naïve native prey may be unable to recognize predators or to respond with sufficient effectiveness (Banks & Dickman, 2007; Kovacs et al., 2012). Furthermore, the inability of naïve prey to recognize and avoid alien predators may improve the ability of alien predators to hunt on these naïve prey (Sih et al., 2010). If alien predators had coevolved with similar prey as the naïve ones in their original habitats, the situation could be more dramatic and even result in the extinction of the naïve prey population (Sih et al., 2010; Blackburn et al., 2004).

As it relates to biological invasions, the case of the reptiles of the Balearic Islands (Spain; Fig. 1) is probably the most typical within the Mediterranean basin, and maybe one of the most concerning worldwide (Silva-Rocha et al., 2015). Nowadays, there are nineteen alien reptiles and only two native, the Balearic lizard (Podarcis lilfordi) and the Ibiza wall lizard (Podarcis pityusensis; Silva-Rocha et al., 2015). We focus the study on Menorca (Fig. 1), where it is probable that the introduction of alien predators by humans in ancient times led to the extinction of the Balearic lizard on the main island (Pérez-Mellado, 2009; Cooper & Pérez-Mellado, 2012). Thus, the sole native reptile of Menorca, the Balearic lizard, only survives today in the surrounding islets, while other introduced lizards have been able to live in the main island of Menorca (Pérez-Mellado, 2009; Salvador, 2014a). The non-native lizards are the Italian wall lizard, Podarcis siculus, with well-established populations all over the island (Salvador, 2014b), and the Moroccan rock lizard, Scelarcis perspicillata, with some restricted populations (Perera, 2002; Perera, 2014). There are also three species of snakes in Menorca, all of them introduced by humans: the viperine snake, Natrix maura, the false smooth snake, Macroprotodon mauritanicus, and the ladder snake, Rhinechis scalaris. Islets on which the Balearic lizard is found are snake-free except for the islet of Colom, on where, there is a population of ladder snakes whose interactions with P. lilfordi are unknown (Mejías & Amengual, 2000).

Figure 1 Studied populations, all of them from Menorca (Balearic Islands, Spain), in order of increasing predation pressure: (1) Podarcis lilfordi of Colom islet, (2) P. lilfordi of Aire islet, (3) P. lilfordi of Binicodrell islet, (4) Scelarcis perspicillata of Lithica, and (5) Podarcis siculus of Canutells.

Note that populations 1, 2 and 3 live in coastal islets while populations 4 and 5 inhabit the main island of Menorca. Blank map obtained from d-maps: http://www.d-maps.com/pays.php?num_pay=270&lang=en.

It is well-known that the Balearic lizard has evolved insular tameness, which is probably extreme and irreversible (Perez-Mellado, Corti & Lo Cascio, 1997; Cooper, Pérez-Mellado & Vitt, 2004; Cooper, Hawlena & Pérez-Mellado, 2009; Cooper & Pérez-Mellado, 2012). Taking advantage of this unique situation, we aim to deepen in the knowledge of insular tameness, studying the ability of lizards to recognize the scent of predatory snakes. It is possible, as with escape behaviour (Cooper, Hawlena & Pérez-Mellado, 2009), that the intensity of antipredatory reactions of lizards to the scent of snakes is related to predation pressure. Therefore, we experimentally assessed the level of response of the three lacertid lizards of Menorca to the chemical cues of their potential predatory snakes. Thus, we hypothesized that P. lilfordi might have lost the ability to recognize the scents of potential predators, being the tamest population. Meanwhile, S. perspicillata, which experiences a middle level of snakes’ predation pressure, is expected to recognize and react to the scents of the saurophagous snake, at least to some level. Finally, P. siculus would be the wariest population, and fully recognize the chemical cues of the predator snake and react with antipredatory behaviours, circumstance that would help this species to colonize and extensively inhabit the main island of Menorca. Thus, if the response to chemical cues of predators reflects the predation pressure level of each population, we would expect the following order of intensity in the reactions of lizards: P. lilfordi Colom < P. lilfordi Aire < P. lilfordi Binicodrell < S. perspicillata < P. siculus.

Methods

Study system

We studied chemoreception of the three species of lizards of Menorca (Balearic Islands, Fig. 1): one is the native lizard of the island (P. lilfordi) and the other two are introduced species (S. perspicillata and P. siculus).

The Balearic lizard, Podarcis lilfordi (Günther 1874) is endemic to Menorca, Mallorca and the Cabrera archipelago (Balearic Islands, Spain). We studied three subspecies: (1) P. lilfordi brauni from Colom island, which coexists with Rhinechis scalaris, (2) P. lilfordi lilfordi from the snake-free island of Aire, and (3) P. lilfordi codrellensis from the snake-free island of Binicodrell. The three islands are close to the coast of Menorca, and show different levels of predation pressure (considering all types of predators): Colom < Aire < Binicodrell (Cooper & Pérez-Mellado, 2012).

The Moroccan rock lizard, Scelarcis perspicillata (Duméril and Bribon 1839), is native to the mountainous region of Morocco and Algeria, with introduced populations in Menorca (Perera, 2014). The studied population lives in the Pedreres de S’Hostal or Lithica, a limestone quarry with high walls (Perera, 2002; Vitt et al., 2002). The population of the Moroccan rock lizard living in Menorca probably belongs to the subspecies S. perspicillata perspicillata (Perera et al., 2007).

The Italian wall lizard, Podarcis siculus (Rafinesque-Schmalz 1810), is native to Italy (continental Italy, Sardinia, Sicily and several coastal islets), Corsica (France) and the east coast of the Adriatic Sea (Salvador, 2014b). However, P. siculus has been introduced to many Mediterranean countries and the United States (Salvador, 2014b). Here we studied the lizard population of Menorca (Balearic Islands, Spain), probably introduced from Sicily and/or Sardinia (Silva-Rocha, Salvi & Carretero, 2012).

Regarding snakes, we used the three species present in Menorca, all of them introduced by humans (Silva-Rocha et al., 2015). The first one is Macroprotodon mauritanicus, which inhabits the main island of Menorca and is known to predate on lizards (Pleguezuelos, 2014). The second one is Rhinechis scalaris, which inhabits the main island of Menorca and the islet of Colom, and is a generalist that predates on endotherms, being proposed as a potential predator of lizards (Pleguezuelos, 1998). The third snake is Natrix maura, which inhabits the main island of Menorca and feeds on aquatic prey, and is certainly not a predator of lizards (Santos, 2014).

Experimental design

We conducted five experiments of discrimination of scents by lizards during the summer of 2013 in Menorca (Spain). We captured 24 lizards (12 males and 12 females) of each population (24 × 5 = 120): (1) P. lilfordi of Colom (mean SVL ± SE: males = 72.42 ± 0.95 mm, females = 66.96 ± 0.71 mm), (2) P. lilfordi of Aire (males = 75.50 ± 1.12 mm, females = 69.12 ± 0.83 mm), (3) P. lilfordi of Binicodrell (males = 69.54 ± 0.41 mm, females = 65.85 ± 0.34 mm), (4) P. siculus, (males = 71.00 ± 1.09 mm, females = 68.81 ± 0.88 mm) and (5) S. perspicillata (males = 51.08 ± 0.45 mm, females = 48.25 ± 0.52 mm). We captured all lizards by noosing and immediately placed them into individual cloth bags inside individual terraria for transport to the laboratory for each experiment. Snakes were captured in Menorca, and were transported in a different vehicle than lizards in order to avoid any odour mixture. All experiments were conducted in the same laboratory in Es Castell (Menorca, Spain). There, we kept lizards inside individual terraria (40 × 25 × 30 cm) in a maintenance room, with a substrate of artificial grass. We fed lizards daily with crickets and Tenebrio molitor larvae, and provided them water ad libitum. We housed snakes in a different maintenance room, also inside individual terraria (50 × 30 × 30 cm) with a substrate of artificial grass and water ad libitum.

Our experimental protocol is similar to the one used in Mencía, Ortega & Pérez-Mellado (2016). It consists in quantifying the behaviour of lizards in terraria with four different scent treatments: ‘odourless control,’ ‘Natrix,’ ‘Rhinechis,’ and ‘Macroprotodon.’ The treatment of Natrix is used as a pungent odour (see Mencía, Ortega & Pérez-Mellado, 2016). We used the same protocol and treatments for the five experiments.

We placed absorbent paper on the floor of every experimental terrarium (60 × 40 × 40 cm) in order to absorb the odour of each treatment. Then, we used the different snakes as odour-donors to impregnate the absorbent paper with their scents, except for the ‘odourless control’ terrarium. These odour-donors were adult individuals of each species. The snakes were placed into the corresponding terrarium 24 h before the beginning of the experiment, and were placed inside them once again during the extra time between trials of other treatments, closing the occlusive plastic cover of the terrarium to avoid odour loss. Each snake was removed from its terrarium two minutes before an experiment trial and re-introduced there after the trial.

Each lizard was subjected once to each treatment following a random order of permutations, resulting in 480 trials (24 lizards × 4 treatments × 5 experiments). Each lizard was tested once a day within their normal activity period (08.00–17.00 h GMT). The experimental room was dark and only the terrarium was illuminated by a 75 W bulb 50 cm above it, providing homogeneous lighting. We maintained a homogeneous constant temperature of 30 °C in the experimental room in order to avoid possible variations in the behaviour of lizards due to temperature. We drew six equal sectors (in two rows by three columns) in the transparent surface of each terrarium in order to count the number of times that lizards moved among sectors. Each trial begun by introducing the lizard into the experimental terrarium, closing the terrarium with the occlusive transparent cover in order to avoid scent loss, and beginning to record its behaviour with a digital recorder for 15 min. Two observers were placed in front of the terrarium, opposite to each other: one observer recorded the behavioural variables with binoculars and the other recorded the number of movements and changes among the sectors of the terrarium. It was not possible to record data blindly because terraria were clearly labelled to avoid mistakes. Additionally, snakes were re-introduced into their terraria after trials. In any case, bias should be low since lizards’ behaviours were registered by the two observers. All specimens remained healthy throughout the study period and we did not detect any signs of stress. Once we finished each of the five experiments, we released all lizards and snakes at their capture sites.

All experiments were performed under the license of the Balear Government (Govern de les Illes Balears, permit CEP 35/2013) and were conducted in compliance with all ethical standards and procedures of the University of Salamanca (Spain).

Behavioural variables

We recorded 16 behavioural variables: (1) ‘Walk latency’: time until the first ‘walk’ movement, (2) ‘Walk’: lizards walk normally, as moving in the wild, (3) ‘Change among sectors’: lizards move between the six predefined sectors of the experimental terrarium, (4) ‘Slow’: lizards walk slowly and with stalking or scattered movements (Thoen, Bauwens & Verheyen, 1986; Mencía, Ortega & Pérez-Mellado, 2016), (5) ‘TF latency’: time until the first TF (tongue-flick), (6) ‘TF’: lizards extrude the tongue and quickly retract it into the mouth, (7) ‘Snout’: lizards tap the wall of the terrarium with the snout, (8) ‘Rubbing’: lizards rub the head against the walls of the terrarium, (9) ‘Stand and scratching’: lizards stand up against the wall of the terrarium and scratch with the forelegs, (10) ‘Head bob’: lizards shake the head up and down, (11) ‘Head raise’: lizards raise the head with the forelimbs straightened, (12) ‘Tail waving’: lizards wave the tail in a horizontal plane, (13) ‘Foot shake’: lizards move the forelimbs rapidly up and down, (14) ‘Walk time’: total amount of time that lizards move normally, (15) ‘Slow time’: total amount of time that lizards move in slow motion, and (16) ‘No move’: total amount of time that lizards stay immobile. The variables were quantified as frequencies, except for ‘Walk latency,’ ‘TF latency,’ ‘Walk time,’ ‘Slow time,’ and “No move,’ which were quantified as duration measured in seconds. We started to record the behaviour of each lizard 5 s after placing it in the centre of the experimental terrarium.

Data analysis

We conducted all analyses on R, version 3.1.3 (R Core Team, 2015). Because neither the original nor log-transformed data met the requirements of parametric statistics, we analysed the data with non-parametric tests. For each experiment, we used the repeated measures Friedman’s test to assess possible differences in the behavioural variables among treatments. That is, we conducted a Friedman’s test on each variable of each experiment with lizard as subject and treatment as grouping factor. When Friedman’s test was significant, we performed post-hoc multiple comparisons for Friedman’s test in order to locate the differences between treatments (Giraudoux, 2012).

Results

Within each studied population, results were similar for males and females on each behavioural variable (Mann–Whitney’s U test, P > 0.05 in all cases), so we pooled data of both sexes within each experiment.

There were no significant differences for any variable in the three studied populations of the Balearic lizard (Tables 1, 2 and 3). Thus, P. lilfordi lizards did not recognize the odour of the potential predatory snakes of Menorca. There were no significant differences between the four treatments for all variables observed in the Moroccan rock lizard either (Table 4). Therefore, S. perspicillata lizards did not recognize the odour of the potential snake predators of Menorca.

Table 1 Mean (range) values of each behavioural variable and results of the Friedman test (df= 3) for the experiment of Podarcis lilfordi of Aire Island (Menorca, Spain) with the four scent treatments (Macroprotodon mauritanicus, Rhinechis scalaris, Natrix maura, and an odourless control).

Only the behavioural variables that were displayed by lizards during the experiments are included in the table.

Variable	Control	Natrix	Rhinechis	Macropr.	Chi-squared	P	
Walk latency	43.25 (2–176)	32.75 (2–142)	38.96 (2–118)	40.79 (4–122)	2.2689	0.5185	
Walk	86.13 (38–138)	82.50 (41–161)	84.67 (23–184)	82.29 (37–123)	0.6176	0.8924	
Ch. sectors	35.21 (10–60)	34.13 (13–72)	38.00 (6–82)	31.29 (17–43)	4.6891	0.196	
TF latency	32.04 (2–160)	30.75 (5–147)	28.50 (5–65)	31.25 (2–120)	0.5443	0.9091	
TF	72.75 (30–126)	74.08 (40–133)	74.79 (5–134)	69.33 (32–140)	1.15	0.765	
Snout	35.62 (9–85)	33.96 (2–93)	37.21 (5–72)	30.00 (13–68)	6.383	0.0944	
Rubbing	50.83 (3–103)	51.87 (6–108)	52.58 (8–124)	46.63 (9–100)	1.4184	0.7012	
Stand and scr.	32.08 (0–133)	36.33 (5–118)	33.33 (3–103)	31.54 (5–93)	1.6835	0.6406	
Head raise	32.21 (16–62)	32.38 (7–60)	38.50 (8–67)	31.75 (14–57)	5.1519	0.161	
Walk time	273.58 (92–448)	283.04 (88–481)	296.67 (64–553)	260.46 (145–440)	2.05	0.5621	
No move	620.42 (452–808)	617.04 (419–812)	621.67 (415–836)	639.54 (460–755)	1.35	0.7173	

Table 2 Mean (range) values of each behavioural variable and results of the Friedman test (df= 3) for the experiment of Podarcis lilfordi of Binicodrell Island (Menorca, Spain) with the four scent treatments (Macroprotodon mauritanicus, Rhinechis scalaris, Natrix maura, and an odourless control).

Only the behavioural variables that were displayed by lizards during the experiments are included in the table.

Variable	Control	Natrix	Rhinechis	Macropr.	Chi-squared	P	
Walk latency	52.92 (5–128)	61.04 (10–180)	62.83 (3–368)	48.00 (6–152)	2.8109	0.4217	
Walk	70.83 (32–220)	68.83 (22–141)	69.46 (9–185)	79.04 (16–153)	2.8243	0.4195	
Ch. sectors	28.63 (14–77)	28.54 (8–56)	30.79 (5–89)	34.63 (8–63)	2.4684	0.481	
TF latency	30.62 (2–120)	35.67 (1–129)	42.67 (3–181)	39.08 (4–165)	3.153	0.3689	
TF	66.25 (29–126)	69.37 (31–110)	58.13 (18–121)	69.96 (27–130)	4.1667	0.244	
Snout	20.96 (1–83)	21.46 (0–54)	22.75 (4–53)	29.00 (6–59)	6.5696	0.0870	
Rubbing	24.67 (2–137)	22.33 (0–45)	22.00 (2–62)	32.58 (1–111)	6.2436	0.1003	
Stand and scr.	17.08 (0–109)	15.33 (0–50)	12.92 (0–58)	10.58 (0–31)	2.6533	0.4482	
Foot shake	0.04 (0–1)	0.17 (0–2)	0.00 (0–0)	0.04 (0–1)	6.1304	0.1054	
Head raise	15.87 (2–43)	16.04 (1–39)	16.08 (0–49)	18.50 (4–53)	1.8155	0.6116	
Walk time	183.88 (65–648)	177.63 (42–380)	117.96 (19–342)	205.08 (43–378)	2.3473	0.5035	
No move	715.25 (252–835)	726.54 (520–858)	722.83 (557–887)	694.87 (522–856)	2.8745	0.4114	

Table 3 Mean (range) values of each behavioural variable and results of the Friedman test (df= 3) for the experiment of Podarcis lilfordi of Colom Island (Menorca, Spain) with the four scent treatments (Macroprotodon mauritanicus, Rhinechis scalaris, Natrix maura, and an odourless control).

Only the behavioural variables that were displayed by lizards during the experiments are included in the table.

Variable	Control	Natrix	Rhinechis	Macropr.	Chi-squared	P	
Walk latency	25.21 (8–57)	35.96 (8–151)	33.63 (7–103)	32.33 (9–86)	3.7089	0.2947	
Walk	93.37 (38–206)	97.83 (53–157)	90.21 (36–181)	91.63 (59–158)	2.2911	0.5142	
Ch. sectors	39.92 (16–72)	43.87 (17–96)	36.62 (12–68)	40.21 (19–61)	0.5083	0.9175	
TF latency	24.50 (2–71)	31.50 (2–130)	28.17 (1–123)	30.58 (3–115)	2.1319	0.5455	
TF	89.79 (34–138)	92.96 (46–172)	84.42 (39–133)	79.08 (35–137)	6.0378	0.1098	
Snout	34.08 (8–59)	42.75 (13–78)	38.29 (3–58)	37.75 (16–69)	5.882	0.1206	
Rubbing	53.21 (3–123)	51.83 (7–93)	49.08 (4–74)	56.96 (23–128)	3.6933	0.2965	
Stand and scr.	37.75 (0–167)	40.96 (0–173)	41.96 (0–177)	41.88 (0–173)	6.4805	0.09043	
Head raise	34.08 (16–59)	36.21 (12–72)	35.58 (17–63)	33.04 (20–49)	0.4635	0.9268	
Walk time	271.79 (106–485)	282.71 (112–454)	276.50 (106–472)	281.96 (140–543)	0.2161	0.9749	
No move	641.12 (415–794)	609.63 (446–748)	626.17 (428–794)	622.63 (357–760)	1.0551	0.7879	

Table 4 Mean (range) values of each behavioural variable and results of the Friedman test (df= 3) for the experiment of Scelarcis perspicillata of the limestone quarry of Pedreres de S’Hostal (Menorca, Spain) with the four scent treatments (Macroprotodon mauritanicus, Rhinechis scalaris, Natrix maura, and an odourless control).

Only the behavioural variables that were displayed by lizards during the experiments are included in the table.

Variable	Control	Natrix	Rhinechis	Macropr.	Chi-squared	P	
Walk latency	50.25 (4–315)	75.08 (3–501)	63.96 (3–390)	109.79 (9–464)	4.85	0.1831	
Walk	20.79 (4–46)	24.71 (8–77)	23.17 (4–72)	16.08 (3–33)	3.4068	0.3331	
Ch. sectors	8.83 (3–27)	10.38 (2–29)	9.96 (2–26)	7.25 (2–16)	2.1261	0.5467	
TF latency	57.92 (5–318)	97.75 (4–495)	61.83 (4–215)	119.67 (9–468)	4.5504	0.2078	
TF	11.04 (2–31)	14.58 (5–43)	11.13 (2–24)	12.17 (3–46)	1.0474	0.7898	
Snout	3.87 (0–8)	5.04 (1–13)	3.63 (0–14)	4.42 (0–26)	6.9324	0.0741	
Rubbing	6.75 (0–37)	11.04 (0–97)	6.46 (0–45)	4.04 (0–21)	5.2043	0.1574	
Stand and scr.	8.04 (0–44)	12.71 (0–81)	9.00 (0–31)	7.25 (0–22)	1.5	0.6823	
Foot shake	0.00 (0–0)	0.29 (0–7)	0.00 (0–0)	0.00 (0–0)	3.0	0.3916	
Head raise	12.50 (3–35)	14.67 (3–41)	14.54 (2–47)	9.92 (0–23)	3.4378	0.3289	
Walk time	83.79 (10–257)	99.83 (22–293)	82.17 (15–235)	62.38 (15–124)	2.4477	0.4848	
No move	816.21 (643–890)	800.17 (607–878)	817.63 (665–885)	837.63 (776–885)	2.7238	0.4362	

However, we found significant differences for all behavioural variables in the Italian wall lizard, except for ‘rubbing’ and ‘TF’ (Table 5). ‘Walk latency’ was higher for lizards in the treatment of Macroprotodon than for the rest, being similar for Rhinechis, Natrix and the odourless control (Fig. 2). In addition, the time moving normally, ‘walk time,’ was significantly lower for the treatment of Macroprotodon than for the other scents, being similar between Rhinechis, Natrix and the odourless control (Fig. 3). Similar results for other variables confirmed that P. siculus lizards recognized the scent of the predatory snake M. mauritanicus, and reacted with antipredatory behaviours, while results indicate that they did not recognize R. scalaris as a predator (Table 6).

Table 5 Mean (range) values of each behavioural variable and results of the Friedman test (df= 3) for the experiment of Podarcis siculus of Es Canutells (Menorca, Spain) with the four scent treatments (Macroprotodon mauritanicus, Rhinechis scalaris, Natrix maura, and an odourless control).

Only the behavioural variables that were displayed by lizards during the experiments are included in the table. Significant differences between treatments are marked in bold.

Variable	Control	Natrix	Rhinechis	Macropr.	Chi-squared	P	
Walk latency	59.21 (5–240)	53.67 (4–175)	53.46 (5–225)	143.21 (25–352)	19.4496	0.0002	
Walk	102.79 (51–213)	111.33 (24–175)	92.87 (8–206)	28.75 (0–71)	41.0084	<0.0001	
Ch. sectors	37.96 (8–60)	38.58 (6–63)	35.54 (4–93)	10.75 (0–28)	39.8787	<0.0001	
Slow	0.04 (0–1)	0.21 (0–5)	5.21 (0–24)	34.33 (11–64)	60.3481	<0.0001	
TF latency	48.83 (2–192)	43.75 (2–172)	49.13 (3–228)	134.04 (19–340)	16.4059	0.0009	
TF	95.50 (39–194)	90.75 (43–228)	95.13 (35–224)	92.50 (31–176)	2.1176	0.5484	
Snout	29.88 (8–60)	29.92 (3–68)	28.33 (0–50)	17.37 (3–34)	24.6992	<0.0001	
Rubbing	78.08 (25–202)	83.04 (7–160)	75.75 (2–268)	64.42 (3–164)	4.9833	0.173	
Stand and scr.	19.71 (0–67)	17.67 (3–54)	15.13 (0–45)	7.04 (0–23)	17.9873	0.0004	
Head bob	0.00 (0–0)	0.00 (0–0)	0.00 (0–0)	0.21 (0–3)	9.0	0.0293	
Head raise	38.71 (10–75)	35.42 (6–83)	34.46 (8–72)	24.00 (0–57)	22.4359	<0.0001	
Tail waving	0.00 (0–0)	0.00 (0–0)	0.13 (0–3)	3.42 (0–17)	37.6667	<0.0001	
Walk time	320.83 (147–696)	323.42 (82–522)	290.17 (45–608)	97.00 (0–284)	33.65	<0.0001	
Slow time	0.13 (0–3)	0.00 (0–0)	14.71 (0–76)	115.38 (30–245)	63.5085	<0.0001	
No move	576.00 (204–750)	576.17 (378–818)	591.21 (178–810)	692.21 (455–827)	19.1849	0.0002	

Figure 2 Boxplots of the time to the first normal movement, or ‘walk latency’, in seconds, of Podarcis siculus lizards of Es Canutells (Menorca, Spain) for the four treatments (Macroprotodon mauritanicus, Rhinechis scalaris, Natrix maura, and the odourless control).

Figure 3 Boxplots of the absolute frequencies of normal movements, or ‘walk’, in number of movements, of Podarcis siculus lizards of Es Canutells (Menorca, Spain) for the four treatments (Macroprotodon mauritanicus, Rhinechis scalaris, Natrix maura, and the odourless control).

Table 6 Observed values of Friedman’s post-hoc paired comparisons of Friedman’s test for the behavioural variables in which differences between treatments were detected for the experiment of Podarcis siculus of Es Canutells (Menorca, Spain) with the four scent treatments (Macroprotodon mauritanicus, Rhinechis scalaris, Natrix maura, and an odourless control).

The critical value of Friedman’s post-hoc comparisons is 29.59 for α = 0.05. Significant differences are marked in bold.

Variable	Control– Macropr.	Control– Natrix	Control– Rhinechis	Macropr.– Natrix	Macropr.– Rhinechis	Natrix– Rhinechis	
Walk latency	31.0	2.5	0.5	33.5	31.5	2.0	
Walk	49.5	0.0	16.5	49.5	33.0	16.5	
Changes among sectors	44.5	7.5	9.0	52.0	35.5	16.5	
Slow	52.5	0.5	17.0	52.0	35.5	16.5	
TF latency	27.5	4.0	1.5	31.5	29.0	2.5	
Snout	38.0	0.5	7.5	37.5	30.5	7.0	
Stand and scratching	31.5	2.0	0.5	29.5	31.5	1.5	
Head raise	40	10	20	30	20	10	
Tail waving	26.5	0.0	1.5	26.5	25.0	1.5	
Walk time	46	3	11	43	35	8	
Slow time	52.5	1.0	16.5	53.5	36.0	17.5	
No move	37	8	13	29	24	5	

Discussion

The results of the native lizard, P. lilfordi, were conclusive. The three studied populations behaved similarly with the scent of potential predatory snakes, with Natrix and with the odourless control, regardless of the predation pressure level of the population. Therefore, two explanations arise. One is that the three studied populations of P. lilfordi have lost the ability to recognize potential snake predators by their scents. The other possible explanation is that the capacity to react to the scent of predators with antipredatory behaviours is a secondary adaptation, and P. lilfordi never had this ability, even when it was crucial for their survival with the introduction of alien predators, to the point that they eventually disappeared from the main island.

The results of non-native lizards living on the main island of Menorca were unexpected. The first one, S. perspicillata, also lacked the ability to recognize scents of snakes, despite sharing its entire distributional range of Menorca with the two snakes, R. scalaris and M. mauritanicus. The other non-native lizard, P. siculus, showed opposite results, recognizing the scent of the potential predatory snake (M. mauritanicus) and responding with clear antipredatory behaviours, such as moving in slow motion and waving the tail. Rhinechis scalaris did not elicit a response in any of the three species of Menorca, reinforcing the previous observations that this snake does not predate on lizards (Valverde, 1967).

We proved that the Balearic lizard, P. lilfordi, does not recognize scents of predatory snakes, regardless of the predation pressure of the population (Binicorell > Aire > Colom; Cooper & Pérez-Mellado, 2012). These results reinforce the strong island tameness to which this lacertid lizard has evolved in the absent of predators (Perez-Mellado, Corti & Lo Cascio, 1997; Cooper, Hawlena & Pérez-Mellado, 2009). The Balearic lizard shows a reduced ability for tail autotomy in comparison with continental Podarcis and even with its sister species, P. pityusensis (Cooper, Pérez-Mellado & Vitt, 2004). Independence of tail autotomy from predation pressure in P. lilfordi suggests that the lack of most antipredatory responses due to island tameness may be fixed for this species, contrarily to what happens in P. pityusensis (Cooper & Pérez-Mellado, 2012; Ortega, Mencía & Pérez-Mellado, 2017). The difference in insular tameness for the two sister species would be attributable to the background risk level of predation that both species have experienced along their evolutionary history since they got separated, more than 2 Ma (Brown et al., 2008; Cooper & Pérez-Mellado, 2012). Podarcis lilfordi would have evolved with a very relaxed predation pressure, free of terrestrial predators and with few birds of prey in Menorca and Mallorca, while P. pityusensis would have evolved under a greater predation pressure in Ibiza and Formentera (Cooper & Pérez-Mellado, 2012). This circumstance would lead the Balearic lizard be so tame that it would have lost the ability to avoid alien predators once humans introduced them to Menorca, more than 5,000 years ago, becoming extinct on the main island and only surviving on the surrounding islets, while P. pityusensis survived the invasive predators (Pérez-Mellado, 2009; Cooper & Pérez-Mellado, 2012; Ortega, Mencía & Pérez-Mellado, 2017).

The Moroccan rock lizard, Scelarcis perspicillata was introduced in Menorca from the North of Africa, probably at various times from the XII century forward (Pérez-Mellado, 2009; Perera, 2014). Although it lives in other areas on the main island of Menorca, its most dense population lives in the limestone quarry of Lithica (Perera, 2002; Perera, 2014). Predation of S. perspicillata by M. mauritanicus was cited in the studied area (Vitt et al., 2002). Even though, Moroccan rock lizards lack the ability to recognize the scent of the potential predatory snake. It is possible that living within the vertical walls of Lithica would make it difficult for snake predators to access, reducing the predation pressure (Vitt et al., 2002; Pérez-Mellado, 2009). In fact, there would be a benefit regarding predators’ avoidance for Moroccan rock lizards living in high perches (mean perch height 78.54 cm, Ortega, Mencía & Pérez-Mellado, 2016), which would entail higher costs than ground habitats regarding the availability of trophic resources (Vitt et al., 2002). It is possible that Moroccan lizards had experienced a reduction of their antipredatory defenses since their arrival to Menorca. In any case, the future study of the African populations that coexist with predatory snakes will reveal whether these lizards have also experienced island tameness or, on the contrary, they never had the ability to recognize scents of predators.

Relevant questions arise from these results: (1) what is the level of background predation pressure necessary to maintain chemical recognition of potential predators; (2) how fast island tameness evolves regarding predator recognition; (3) is there a relationship between predation pressure level and the rate of loss of antipredatory behaviour; and (4) under what circumstances can island tameness become irreversible? These questions and many more are still unanswered about the loss of antipredatory adaptations and its causes and implications for native animals under invasions. It is probable that the rate of loss of any antipredatory defense would be related to its cost in a predator-free environment (Van Damme & Castilla, 1996; Blumstein & Daniel, 2005; Rödl et al., 2007). This would be the situation in islands with scarce trophic resources, as the surrounding islets of Menorca for P. lilfordi or the vertical rock surfaces of Lithica for S. perspicilla. It is hypothesized that the loss of wariness would lead lizards to develop the time-required trophic behaviours that P. lilfordi does in islets as Aire, sucking the nectar of flowers in a highly exposed way (Cooper & Peréz-Mellado, 2004; Pérez-Cembranos, Pérez-Mellado & Cooper, 2013).

The other non-native lizard of Menorca, the Italian wall lizard, P. siculus, has the ability to recognize the chemical cues of M. mauritanicus and respond with antipredatory behaviours. These antipredatory responses include behaviours that would avoid predator detection, such as walking in slow motion, others that suggest vigilance, such as raising the head, and others that would distract attention from vital parts of the body, such as tail vibration (Avery, 1991; Thoen, Bauwens & Verheyen, 1986; Mencía, Ortega & Pérez-Mellado, 2016). Italian wall lizards do not recognize R. scalaris as a predatory snake, which reinforces the evidences that the ladder snake does not predate on lizards. The extraordinary effectiveness of P. siculus clearly detecting the predatory snake and ignoring other non-dangerous snakes would be related to the great adaptability of this species to different environments, which would be related to its good colonizing capacity (Silva-Rocha et al., 2014). Italian wall lizards recognize chemical cues of the predatory snake Hierophis viridiflavus in Corsica, and react with similar antipredatory responses, which include slow motion and tail vibration (Van Damme & Quick, 2001). In addition, P. siculus of Corsica modifies the use of microhabitats in the presence of snake scent (Van Damme & Quick, 2001). Nonetheless, these adaptations would be extraordinarily flexible in the Italian wall lizard, as illustrated with the evolution of insular tameness only 30 years after the introduction of a population of P. siculus in a predator-free islet of Croatia (Vervust, Grbac & Van Damme, 2007). In fact, the fast phenotypic evolution of the Italian wall lizards may be related to their remarkable ability to colonize new environments (Vervust, Grbac & Van Damme, 2007).

Conclusions

Our study shows that the native lizard, P. lilfordi, lacks the ability to recognize potential snake predator by their scents. This lack of response was similar for the three studied populations, regardless of the predation pressure and the presence or absence of snakes. One species of alien lizard, S. perspicillata, also lacked this skill even if the population coexists with R. scalaris and M. mauritanicus. The possible reason for the lack of response of this lizard is related to their living within vertical walls, where snakes have difficult access. The other introduced lizard, P. siculus, recognizes the chemical cues of the potential predatory snake (M. mauritanicus) and responds to them with typical antipredatory behaviours. Regarding the ladder snake, R. scalaris, it did not elicit a response in any of the three lizards, which supports the idea that it does not predate on lizards. Our results rise questions about the loss of chemical recognition of predators during island tameness and its related costs and benefits for lizards of insular habitats. In any case, the extreme insular tameness of P. lilfordi invites to further research on the evolution of antipredatory defenses in lizards and strong conservation measures to avoid invasion of their habitats by alien predators.

Supplemental Information

Data S1 Raw data of all the variables (see manuscript) for the five studied populations of lizards

The number of the lizard identify each individual within each population.

Click here for additional data file.

We thank Mary Trini Mencía and Joseph McIntyre for linguistic revision, and Mario Garrido, Ana Pérez-Cembranos, Gonzalo Rodríguez and Alicia León for assistance capturing lizards and support during writing. Finally, we thank Elisa Soteras for kindly housing us during part of the fieldwork and the staff of Lithica for allowing and facilitating the fieldwork in the quarry.

Additional Information and Declarations

Competing Interests

Author Contributions

Animal Ethics

Field Study Permissions

Data Availability

The authors declare there are no competing interests.

Abraham Mencía performed the experiments, analyzed the data, reviewed drafts of the paper.

Zaida Ortega performed the experiments, analyzed the data, wrote the paper, prepared figures and/or tables, reviewed drafts of the paper.

Valentín Pérez-Mellado conceived and designed the experiments, analyzed the data, wrote the paper.

The following information was supplied relating to ethical approvals (i.e., approving body and any reference numbers):

All experiments were conducted in compliance with all ethical standards and procedures of the University of Salamanca (Spain).

The following information was supplied relating to field study approvals (i.e., approving body and any reference numbers):

All experiments were performed under the license of the Balear Government (Govern de les Illes Balears, permit CEP 35/2013).

The following information was supplied regarding data availability:

The raw data has been supplied as Data S1.

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
