# Peer review of "From tameness to wariness: chemical recognition of snake predators by lizards in a Mediterranean island"

_PeerJ, doi:10.7717/peerj.2828_

## Round 0.1 · original submission · Minor Revisions

· Academic Editor

Minor Revisions

Dear authors,

Our reviewers had some concern about your ms, but I believe that you can address them.

I look forward to your revision

Regards

Michael Wink
Academic editor

Reviewer 1 ·

Basic reporting

This is a straightforward manuscript that explores the specific and population differences in detection of snake predators by insular lizard species subjected to different levels of pressure by predation. The results are clear and surprising. Only the exotic P. siculus is able to detect the presence of saurophagous snakes, showing it by significantly different measurable behaviours from the remaining species and populations tested in the study. Whereas S. perspicillata, an alien lizard species naturally evolved in syntopy with snake predators, apparently did not show any significant behaviour that demonstrated in this lizard species the detection of different saurophagous and non-saurophagous snake species. The authors explain this result attending to the climbing habits of S. perspicillata that may confer an advantage to this lizard species that may avoid the costs related to escaping or hiding behaviours. Finally, the third lizard species studied, P. lilfordi, was studied in three coastal islets from Menorca. The native lizard species did not show any evidence that could be indicating the ability to detect snake predators. The authors suggest that this tameness shown by the native species, typically found in island species, might influence on the extinction of this lizard in the main island when invasive snakes were introduced.

I believe that the manuscript is overall well written. The introduction makes its role, it introduces correctly the objective of the study and the references are up to date. However, I have some points that should be clarified before this manuscript could be acceptable for its publication.

Experimental design

My main concern is in relation to the data analysis. Did the authors consider using a GLMM including the distribution of the dependent variables in the models and including the species and sex as main factors and the population as random factor?
If this is not of convenience of the authors, at least explain better and with more detail what variables were included in each analysis in the Friedman’s tests (i.e. treatment as grouping factor, etc)

Validity of the findings

Lines 233-234. Authors declare that P.lilfordi lose their ability to recognize potential snake predators to the light of their results. However this is a strong phrase. Why did they lose this ability? Maybe this is a secondary adaptation and these lizards never had this ability. What is surprising to me is that they didn’t show or developed this ability even when it was crucial for the survival of the species to the point that they eventually disappeared from the main island. I suggest moderating this phrase. We don’t know if they had this ability before although this may be expected since closely related taxa show it.

Comments for the author

Line 237. “which are living in close syntopy with R. scalaris, probably from hundreds of years” add a reference here if any.
Line 243-245. “Rhinechis scalaris did not elicit a response in any of the three species of Menorca, reinforcing the previous observations that this snake does not predate on lizards.” Add a reference here.
Line 257-258. “have experienced along their evolutionary history since they got separated, more than 2 Ma” add a reference here if any.
Line 262-263. “This circumstance would lead the Balearic lizard be so tame that it would have lost the ability to avoid alien predators once humans arrived in Menorca, more than 5000 years ago”
Maybe “would lead the Balearic lizard to be so tame…”
I understand here that you say lizards lose their ability to avoid alien predators as consequence to the arrival of humans? Humans were the alien predators of lizards (and other fauna) at the beginning?
Lines 275-278. Lizard don’t choose to live in one place or other. They are naturally selected… rephrase these lines. Anyhow, I think this phrase is quite speculative. I would suggest deleting it.
Lines 304. “…clearly avoiding”. I believe it is “clearly detecting” or recognizing. You did not test the ability of lizards to flee away from predators or to avoid being predated.
Line 328-329. As you mentioned P. lilfordi shows reduced tail autotomy and wariness in comparison with close related lizard species. Thus, tail autotomy and chemical recognition of predators are two evolutionary lost abilities that maybe were lost simultaneously. Rephrase accordingly. This is a strong sentence as conclusion.
Minor points
Line 20. Saurophagous, not sauriophagous. Check throughout the manuscript.
Lines 22-23. Different order than in line 95.
Line 46-47. Check Curio 1966 (1976 in reference list)
Line 50. Environments
Line 54. Check Simberloff (et al.?) 2013
Lines 55-56. Check Kovacks et al. 2012 (Kovacs et al. 2012 in references)
Lines 88-89. Rephrase for clarity. And “sauriophagous” again.
Line 95. Different order than in Abstract. Check
Line 102. Change “Podarics”
Lines 106-108. This phrase refers only to predation pressure by snakes? Does include other predators (birds, etc)?
Line 149. Check and homogenize “odorless” versus “odourless” throughout the manuscript.
Lines 186-187. Tongue-flick after the first time TF. You wrote it after the second TF.
Lines 190-191. Delete “as if they were trying to scape”
Line 224. Being similar between replacing “being also similar for”?
Lines 225-227. This phrase may be better in Discussion rather than in results.
Line 306. “good” replacing “huge” colonizing…
Line 356. Change Podarcis into Podarcis (with italics)
Table 6. Mark in bold the significant post-hoc results

·

Basic reporting

The research reported follows a clear experimental design and shows a confincing result. I t explains very well why introduction of predators often leads to extinction of endemic species on islands, while introduced prey species survive.

Experimental design

Welldone.

Validity of the findings

Important for nature conservation and evolutionary biology.

Comments for the author

The only snake you did not comment on like the others was Natrix maura. I recommernd that you add some sentences about it (results/discussion).

Some corrections to the manuscript:
Lines 18/19: Saurophagous, not sauriophagous
Line 22/23: Aire and Binicodrell being snake-free islets.
Line 45: environments
Line 50: Kovacs (as in references)
Line 79: tamest, not most tame
Line 101: Vitt, Cooper & Perera: In other cases with three authors you write 'et al.'. Why not here?
Line 154: Each trial was begun
Line 175: straightened
Line 194: no significant differences
Line 213: seem to have lost
Line 286: the predator-free islet of Croatia: Name the islet or write 'a predator-free islet'.
Tables 5 and 6: The bold markings are not visible in my copy.

---

## Round 0.2 · Major Revisions

· Academic Editor

Major Revisions

Dear authors

One of the reviewers is not satisfied with your revision. Therefore, I have to ask you for another round of revision. The critical reviewer will see your revision again.

Greetings
M Wink

Reviewer 1 ·

Basic reporting

-The authors have improved the manuscript following the comments given by previous revision. However, I still find some points that should be discussed and, eventually, revised.

-Lines 242-243. “Therefore, the three studied populations of P. lilfordi seem to have lost the ability to recognize potential snake predator by their scents. This lack of predator recognition was expected for the populations of Aire and Binicodrell, snake-free islets, but is rather surprising for lizards of Colom, which are living in close syntopy with R. scalaris, probably from hundreds of years (Pleguezuelos 1998)”

So, why did you expect (and it was surprising) that P.lilfordi were adapted to recognize a non-saurophagous snake if scent recognition is costly? Rewrite this paragraph.

-Lines 245-246. “even when it was crucial for their survival”

When were the saurophagous snakes introduced in the island? It might be crucial for the survival of lizards in the presence of saurophagous snakes, once they were introduced. But, R. scalaris, supported by your own results (lines 256-257) may not be a saurophagous snake. Delete this sentence.

-Lines 247-248. “In any case, the extreme insular tameness of P. lilfordi invites to further research on the evolution of antipredatory defenses in lizards and strong conservation measures to avoid invasion of their habitats by alien predators”
Maybe I would write this sentence in conclusions.

Experimental design

In M&M section
-I would like to know, and it may be useful for the audience, if observations were done by the same observers that set the scent experiments. Were the observations “blind” in relation to the scent treatment that had been performed in the arenas?

Validity of the findings

-One of my biggest concerns is in relation to the sentence: (lines 32-34) “In general, our results suggest that chemical discrimination might be evolutionarily lost sooner than other antipredatory adaptations, such as tail autotomy or escape behaviour.”

As far as I understand you do not provide any proof that evidenced investment in chemical recognition of predators is more costly than, specifically, tail autotomy. But, if you have references for this, please provide as much as you can.
Indeed you comment in lines 263-265 “The Balearic lizard shows a reduced ability for tail autotomy in comparison with continental Podarcis and even with its sister species, P. pityusensis (Cooper et al. 2004)”. This phrase contradicts the previous lines. Thus, P. lilfordi normally does not autotomize its tail. As far as I understand here, you do not know if this lack of antipredatory defence was a previous or later adaptation to “loss” of scent recognition. Besides, in my opinion, it is hard to believe that scent recognition is evolutionary lost before tail autotomy because it was more costly than autotomizing the tail, which is per se a fat-storage tissue. Overall, is what I understand from lines 32-34 and lines 305-306. If you consider this an accurate comment, please rewrite it to soft your sentences and maybe remove the tail autotomy part.

Comments for the author

-In lines 272-273 you write “Podarcis lilfordi would have evolved with a very relaxed predation pressure, free of terrestrial predators and with few birds of prey in Menorca and Mallorca”

It makes me think that P.lilfordi never had the ability, or lost it long time ago, to respond predators. As I see your study, your results may lead you to write an interesting story about conservation issues in insular environments of native species with naïve behaviours in presence of alien predators. What I understand is that you do this here. But, sometimes you tried to explain evolutionary processes that are hard to believe only based on these experiments. I believe you could do a better work trying to show the conservation issues of native lizard species with reduced abilities to scape invasive predators in islands. In my opinion, that would rise attention of environmental services more than a story about evolution.

-One last suggestion. I recently read that Scelarcis perspicillata was moved into genus Teira. Teira perspicillata. Revise and take a decision based on last phylogenies.

---

## Round 0.3 · accepted · Accept

· Academic Editor

Accept

Dear authors

Good news! Now we can accept your revised manuscript. Thanks for submiiting your work to our journal.

Greetings

Michael Wink
Academic editor

Reviewer 1 ·

Basic reporting

In my opinion, the article is now correctly written

Experimental design

Already commented in previous versions

Validity of the findings

Very interesting from both evolutionary and conservation points of view

Comments for the author

Thank you for following my comments. I think you improved the manuscript considerably and it deserves publication as a good contribution to the area